# SEHIDS: Self Evolving Host-Based Intrusion Detection System for IoT Networks

**DOI:** 10.3390/s22176505

**Published:** 2022-08-29

**Authors:** Mohammed Baz

**Affiliations:** Department of Computer Engineering, College of Computer and Information Technology, Taif University, Taif 21994, Saudi Arabia; mo.baz@tu.edu.sa

**Keywords:** IoT, intrusion detection systems, constructive neural networks

## Abstract

The Internet of Things (IoT) offers unprecedented opportunities to access anything from anywhere and at any time. It is, therefore, not surprising that the IoT acts as a paramount infrastructure for most modern and envisaged systems, including but not limited to smart homes, e-health, and intelligent transportation systems. However, the prevalence of IoT networks and the important role they play in various critical aspects of our lives make them a target for various types of advanced cyberattacks: Dyn attack, BrickerBot, Sonic, Smart Deadbolts, and Silex are just a few examples. Motivated by the need to protect IoT networks, this paper proposes SEHIDS: Self Evolving Host-based Intrusion Detection System. The underlying approach of SEHIDS is to equip each IoT node with a simple Artificial Neural Networks (ANN) architecture and a lightweight mechanism through which an IoT device can train this architecture online and evolves it whenever its performance prediction is degraded. By this means, SEHIDS enables each node to generate the ANN architecture required to detect the threats it faces, which makes SEHIDS suitable for the heterogeneity and turbulence of traffic amongst nodes. Moreover, the gradual evolution of the SEHIDS architecture facilitates retaining it to its near-minimal configurations, which saves the resources required to compute, store, and manipulate the model’s parameters and speeds up the convergence of the model to the zero-classification regions. It is noteworthy that SEHIDS specifies the evolving criteria based on the outcomes of the built-in model’s loss function, which is, in turn, facilitates using SEHIDS to develop the two common types of IDS: signature-based and anomaly-based. Where in the signature-based IDS version, a supervised architecture (i.e., multilayer perceptron architecture) is used to classify different types of attacks, while in the anomaly-based IDS version, an unsupervised architecture (i.e., replicator neuronal network) is used to distinguish benign from malicious traffic. Comprehensive assessments for SEHIDS from different perspectives were conducted with three recent datasets containing a variety of cyberattacks targeting IoT networks: BoT-IoT, TON-IOT, and IoTID20. These results of assessments demonstrate that SEHIDS is able to make accurate predictions of 1 True Positive and is suitable for IoT networks with the order of small fractions of the resources of typical IoT devices.

## 1. Introduction

Accessing anything from anywhere at any time is the ultimate goal that the Internet of Things (IoT) is devised to achieve [1]. It is, therefore, not surprising that the IoT has been integrated into a wide range of applications, including smart homes [2], e-health [3], and intelligent transportation systems [4]. To put things into perspective, studies on IoT statistics showed that there were around 13.8 billion devices in use by the end of 2021 and that this number will double by 2025 [5]. The massive proliferation of IoT networks and the heavy reliance on them to manage various critical aspects of our lives prompt the desperate need to devise effective mechanisms that can protect these networks from malicious access.

Indeed, since the advent of the IoT, it has been the target of several attacks using unprecedented techniques. These include (i) recruiting IoT devices to form wanton botnets, as in the case of the Dyn attack [6], in which about 2.5 million malicious IoT nodes were conscripted to execute a distributed denial of service (DDoS) attack against the Dyn domain name service. The volume of traffic generated by this attack was estimated at 1.5 terabits per second and was capable of bringing down several high-profile websites, including Twitter, The Guardian, and Netflix. (ii) Exploiting the vulnerabilities of IoT devices to destroy them, as in the case of BrickerBot.1 [7] and its successors: BrickerBot.2, BrickerBot.3, and BrickerBot.4 [8], which focus on compromising IoT devices and corrupting their storage to form what is known as a Permanent Denial of Service (PDoS) attack. (iii) The use of electromagnetic waves to attack microelectromechanical subsystems of IoT devices, as in the case of the Sonic attack [9], which uses tuned acoustic tones to deceive accelerometers of IoT devices in such a way that transfers control of the deceived devices to the intruder. In addition to the aforementioned, other attacks aiming to breach IoT privacy have also been reported in the open literature, such as Smart Deadbolts [10] and Silex [11].

Protecting IoT networks from cyberattacks is carried out conventionally by deploying a Network Intrusion Detection System (NIDS) on edge devices such as border routers or access points [12,13]. NIDS scans incoming and outgoing traffic traversing the network with the aim of identifying suspicious actions that threaten one or more security aspects. However, NIDS can provide global protection for the entire network, it has several shortcomings, such as a single point of failure and a lack of ability to detect internal malicious activities. Stimulated by these shortcomings, the host-based IDS (HIDS) was proposed [12,13]. The key concept on which this approach is based is to equip each IoT device with an IDS mechanism so that it can autonomously detect cyberattacks. Various approaches have been used to develop HIDS: statistical analysis [14], pattern recognition [15], and rule matching [16], to name a few. However, the high proficiency of Artificial Intelligence (AI) makes it the de facto approach for the development of most contemporary IDSs [17,18]. Therefore, this study adopts one of the widely used AI approaches: Artificial Neural Networks (ANNs) [19]. The main reason for choosing ANN over other AI approaches, e.g., Support Vector Machine [19], XGBOOST [20], or Random Forest [21], is the wide variety of architectures that can be implemented by ANN. Multilayer Perceptron (MLP), Recurrent Neuron Networks (RNN), Convolutional Neuron Networks (CNN), Graph Neural Networks, and Modular Neural Networks are some popular examples. In addition, ANN can be constructed to carry out various tasks such as regressors, classifiers, replicators (e.g., Replicator Neuron Network (ReNN) and autoencoders), and/or generators (e.g., generative adversarial networks) and can adapt various learning strategies including supervised, semi-supervised and unsupervised. Another important reason for using ANN in this study is its great scalability, which allows multiple shallow networks to be cascaded to form a deeper network. Interestingly, the great competence of ANN is justified by the universal approximation theory [22], which proves the ability of the ANN architecture to approximate an arbitrary dynamical system from a set of observations generated by that system to some degree of accuracy without having to track the internal states of the system. However, the high computational cost that ANNs require, in conjunction with the stringent characteristics of IoT networks, stands against the straightforward deployment of ANN algorithms on these networks.

A typical IoT network consists of a set of scarce resource devices with an ad hoc topology and heterogeneous traffic patterns [1,2,3,4,5]. The resource scarcity of IoT devices is driven by the need to miniaturise these devices to such sizes that make it feasible to embed them into different objects. The ad hoc topology is a direct consequence of the fact that an IoT network is typically deployed to harvest information from a specific physical environment, which presumably has its own unique structure. Finally, the heterogeneity of traffic across an IoT network is due to the co-existence of different types of nodes (e.g., monitoring, alerting, and telemedicine), each of which has its own traffic characteristics. In addition, the short transmission ranges of IoT devices, combined with the fact that the batteries of some devices run out faster than others, increase the need to redistribute traffic on a regular basis. This, in turn, amplifies the heterogeneity and turbulence of traffic amongst nodes.

Encouraged by the benefits of using ANN algorithms in IoT and other resource-constrained networks, several resource thinning techniques have appeared in the open literature, such as transfer learning [23], federated learning [24], and quantisation [25]. The idea of transfer knowledge is to offload the most expensive resource operations (i.e., batch training) to flagship devices thereafter to make the trained model available to run over tiny devices. Transfer learning can save the time and resources required to train the model. However, the lack of a well-representative training dataset for IoT traffic makes it challenging to ensure that the trained model is appropriate for the traffic it will face at deployment (i.e., negative transfer [26]). Another major shortcoming of transfer learning is the significant latency required to train and then transfer the knowledge to each device. Federated learning is another technique that has been proposed to facilitate the deployment of ANN models on tiny resource devices. The core idea of this technique is that each device can train its model locally. After that, models ‘parameters are sent to a designated server whose main task is to aggregate these weights and then disseminate them back to each node to update its weights accordingly. Although such an approach can save a significant number of computational resources by distributing the computational load across multiple devices, the exchange of parameters over the network can squander this saving, as the energy required to transmit a bit is about a hundred times the energy required to process it locally [27]. Quantisation is another simplification approach that aims to reduce the computational complexity of ANN models by scaling down the space required to represent the operands (i.e., the weights and biases) and the operations (i.e., the activation functions of the neurons) and/or even the input to the model. Various scaling strategies have been proposed, ranging from removing the floating part of the quantised numbers to binarisation. The obvious advantages of quantisation are that it can reduce computational costs, memory requirements, and the storage capacity required by the model. However, this comes at the expense of lower prediction accuracy, a slowdown in learning speed and the need to use more training examples [25].

Despite the differences among these thinning techniques, the mainstay method for developing most ANN models is to evaluate the performance of a bunch of them with different configurations using one or more training datasets. Then the model with the best result is put into practice regardless of how well it responds to the real field data. Nevertheless, using a fixed ANN model by different devices whose requirements are different, together with the lack of well-representative training datasets for IoT traffic, can lead to a waste of resources, poor predictions, or even a combination of both.

Notwithstanding, we argue that designing an effective HIPS for IoT networks requires a more agile approach that allows each node to tailor its own ANN architecture in accordance with the dynamics of the received traffic. Inspired by this argument, this paper proposes a novel class of HIDS for IoT networks: Self Evolving Host-based Intrusion Detection System (SEHIDS). Basically, SEHIDS equips each IoT node with a simple ANN architecture consisting of an input and an output layer and a lightweight mechanism through which an IoT device can train this architecture online and evolves it whenever its performance prediction is degraded. SEHIDS meets the stringent characteristics of IoT networks from different perspectives. Firstly, SEHIDS facilitates retaining the ANN architecture to its near-minimal configurations, which can save the resources required to compute, store, and manipulate the model’s parameters as well as the time needed to generate the results. Another significant advantage of SEHIDS in terms of maintaining the near-minimal model is that it speeds up the convergence of the model and avoids the problems associated with the recruitment of dispensable layers in the model, e.g., overfitting and/or exploding and vanishing gradients [19]. Secondly, SEHIDS allows each node to generate the architecture that matches its own traffic characteristics and adapts it accordingly. This is not only compatible with the discrepancy and turbulence of traffic amongst nodes, but it can also widen the applicability of our proposal by eliminating the need to make a priori assumptions about the position of nodes or their role in the network. Finally, SEHIDS eliminates the need for conducting a separate training phase to select the best model by evolving the model incrementally using the data as received, which can speed up the plan-to- the production cycle of our proposal.

Several approaches and techniques are utilised to design SEHIDS. These include devising the conditions under which a new layer is added, specifying the number of neurons in the new layer, developing the pattern used to connect this new layer to the current architecture, and the learning scheme used to train the model. To ensure the tractability of SEHIDS, the constructive neural networks [28] and cascade correlation learning [29] are used as the design framework. While maintaining the learning capacity of SEHIDS in coinciding with the traffic dynamics is carried out by devising novel versatile criteria for the evolution of SEHIDS based on the divergence between the errors produced by the model due to the new input and the residual errors produced by the model so far. To avoid exhausting the resources of the IoT devices by expanding the hypothesis space of SEHIDS beyond what is required, the number of neurons in newly added is adjusted based on the number of remaining parameters to achieve the optimal degrees of freedom. More importantly, the use of these approaches and techniques enables us to decouple the simple ANN architecture that is fed into the SEHIDS in the initial phase from the evolving mechanism. This, in turn, facilitates using SEHIDS with different ANN architectures. In order to take advantage of this capability, this study develops the two common types of IDS: signature-based and anomaly-based. In the signature-based IDS version, a supervised architecture (i.e., MLP [19]) is used to classify different types of attacks, while in the anomaly-based IDS version, an unsupervised architecture (i.e., ReNN [30]) is used to distinguish benign from malicious traffic.

The performance of SEHIDS was comprehensively assessed against three contemporary datasets containing a variety of cyberattacks targeting IoT networks: BoT-IoT [31], TON-IOT [32], and IoTID20 [33]. These assessments are designed to serve two purposes: firstly, evaluating the ability of SEHIDS to make an accurate detection for different types of attacks, and secondly, investigating the ability to run SEHIDS on devices with scarce resources. These two assessments were quantified using various performance metrics, including the standard statistical measurements that are usually used to determine classification accuracy, e.g., True Positive, False Positive, True Negative False Negative, accuracy, recall and F1-score and those metrics that are used to measure the number of resources, e.g., FLoating Point OPerations per Second (FLOPs) and memory usage. The results of these assessments demonstrate the high competence of SEHIDS to fulfil both purposes. Indeed, the average achieved true positive rate is 1. Furthermore, these results show that the resources consumed by SEHIDS are in the order of small fractions of typical IoT devices’ resources.

The rest of this article is structured as follows: Section 2 reviews the most pertinent works presented in the open literature, and Section 3 describes the datasets used to assess the performance of our proposal. The methods used to develop SEHIDS are presented in three subsections: Section 4.1 provides the problem formulation and assumptions underlying SEHIDS, a detailed description of SEHIDS is given in Section 4.2, finally, Section 4.3 discuss how SEHIDS can be used as a signature classifier or anomaly detectors. Section 5 presents the results and discussions, and finally, Section 6 concludes this paper.

## 2. Related Works

Designing an effective Intrusion Detection System (IDS) is one of the long-standing practices that has been carried out using various approaches and techniques. Most studies and systematic reviews deal with the enormous number of proposals by compartmentalising them according to different criteria: for example, according to the location where IDSs are deployed into host-based and network-based, according to the detection strategy into signature-based and anomaly-based, according to the responsiveness to intrusions into active and passive, according to the way by which they are updated into offline and online. As these criteria are neither mutually exclusive nor endless, we concentrate here on the detection strategy, as it plays a paramount role in designing IDSs for IoT networks.

The underlying assumption of signature-based IDS is that an intrusion can be typified by a set of rules and patterns; hence by comparing the ongoing traffic with those that preceded and/or coincided with the occurrence of the well-known intrusions, potential attacks can be identified. The anomaly-based HIDS, on the other hand, is devised on the concept that anomalous traffic deviates from normal behaviour; therefore, intrusions can be figured out just by determining whether a particular traffic instance is normal or not. This, in turn, makes signature-based systems an excellent candidate for timely detection of known attacks with high accuracy. However, this excellent performance can be discredited in the face of unknown attacks such as polymorphic malware and zero-day attacks. Although anomaly-based systems can perform better in these circumstances, the validity of these systems is endangered when they are tricked either maliciously, as is the case with adversarial attacks [34,35,36], or benignly by the emergence of legitimate but never-before-seen traffic profiles due to appearing of new services or protocols [37,38]. The remaining of this section is devoted to reviewing the fundamental concepts used to develop IDS for IoT with the aim of highlighting the merits of our work in comparison to related work.

Employing devices’ resource usage as an indicator of malicious activity is one of the earliest approaches that was used to design network-based intrusion detection systems for resource-constrained networks. The work presented in [39] uses this approach to develop three statistical machine learning models based on linear regression, multilayer perception (MLP) and Recurrent Neural Networks (RNNs) to identify different types of IoT attacks, including unauthorised access, port scanning and viruses. The proposed models poll resources used by each IoT device (e.g., CPU usage, memory usage, storage occupancy) within a predefined time window and then feed them into each of the three models. These models compare the actual usage values with the predicted values they generated at the previous time window to determine the presence of an attack. The results reported in this paper show that the model is able to detect the said attacks just by monitoring the misuse of resources. However, this assessment uses a bespoke dataset generated from a simple network of only 12 BeagleBone Black devices. This makes its performance questionable in more realistic cases. The model presented in [40] follows the same approach as in [39], except that it uses energy consumption in lieu of other resource utilisation readings. This work argues that energy anomaly is an effective indication of attacks and that the type of attack (either physical or cyber) can be detected from the energy profile. Nevertheless, this model uses several pre-processing techniques to reduce the noise that affects the energy values before feeding them into the Deep Learning model. A fixed CNN model of nine convolutional layers with three flat and three dense layers is used to disaggregate different components of the received energy before passing them to another CNN model to make the prediction. The later CNN model consists of two convolutional layers with max-pooling, flat and dense layers.

Despite the high-performance readings reported by the aforementioned works, the main shortcoming in the development of NIDS based on measuring resource misuse is that it cannot be used to detect those types of attacks that do not consume noticeable resources, such as passive attacks or those that do not take much time to compromise devices such as Botnet. To overcome this shortcoming, some work has used traffic analysis to develop NIDS. The work presented in [41] employs a deep learning approach to develop a novel online and anomaly-based NIDS that uses raw data packets as the input. The proposed system, named Kitsune, consists of the following five components: Packet Capture, Packet Parser, Feature Extraction, Feature Mapping and Anomaly Detector. The main task of Packet Capture, as its name implies, is to capture the incoming and outgoing raw packets traversing the monitored network. The output of this module is then passed to Packet Parser, whose main task is to extract the meta information such as arrival time, sequence numbers and packet sizes. Feature Extractor is the third module, which is used to extract the most relevant information from the supplied metadata and then combine it into tuples using the damped incremental statistics algorithm. The Feature Mapper then applies some dimensionality reduction algorithms to transform the supplied data into the shape that matches the dimension of the input layer of the Anomaly Detector component. Finally, the Anomaly Detector, which acts as the main component of Kitsune, consists of two other subcomponents: Ensemble and Output, each of which is defined in a group of three layers based on the MLP autoencoder architecture. During the training phase, Ensemble is used to allow Kitsune to recognise the behaviour of benign traffic, while in the testing phase, Ensemble is used to pass the prediction error to the Output subcomponent. Output is responsible for the final decision on the input packet. Kitsune uses stochastic gradient descent to facilitate online training of the model and adaptive damped windows to limit memory usage without setting a hard threshold. The performance of Kitsune was evaluated using an attack dataset generated from custom-built networks to mimic a smart home environment. The results of this evaluation show that increasing the number of layers in Ensemble reduces detection time and increases classification accuracy. However, these results also show that some types of attacks, such as DoS, cannot be detected by Kitsune at an earlier stage, as a long sequence of operations (parsing, extraction, and mapping) must be performed on numerous packets before DoS behaviour can be identified. Another serious shortcoming of Kitsune is that it must be trained with benign traffic. Failure to do so not only results in significant performance degradation but also leaves Kitsune vulnerable to adversarial machine learning attacks. Interestingly, this vulnerability limits the ability to use Kitsune to secure running networks.

Realguard [42] is an ANN-based Network Intrusion Prevention System (NIPS) that follows the same approach as [41] but has been optimised to reduce computational complexity by combining some of the components of Kitsune. An active manager component has also been added as a means to specify the preventive and/or reporting actions to be taken when intrusive traffic is detected. In addition, Realguard uses the MLP architecture to develop a DNN consisting of five hidden layers with a total of 34,315 parameters without using an autoencoder architecture such as Kitsune. The results of this work show that Realguard can achieve an overall accuracy of 99.57% at a processing rate of 10,600 packets per second on the CIC-IDS2017 dataset using the Raspberry PI as an evaluation environment. Furthermore, a comparison between Realguard and Kitsune shows that Realguard is able to reduce CPU usage and memory occupancy by a factor of 0.91% and 0.73%, respectively, compared to Kitsune. Despite the superior performance of Realgurad, the authors of this paper point out some of its major limitations, such as the lack of a barrier layer to protect Realgurad from adversarial attacks, the need to use an advanced learning approach such as federated learning to enable deployment on multiple edge devices simultaneously, and the significant number of resources required to train the model.

Performing all IDS activities by a network device requires equipping NIDS with several components to handle the high volume of traffic traversing the network and then making a decision about it. This sophisticated structure and the need to deploy multiple NIDS in the network to avoid a single point of failure have led some researchers to delegate some IDS activities to IoT nodes instead of performing them all on NIDS. For example, SVELTE [43] was one of the pioneer’s IDS systems that was designed especially to comply with the processing capabilities of the constrained nodes by offloading the resource-intensive operations to the border router and keeping lightweight operations to be performed locally by each device. SVELTE allows border routers to draw the baselines of the networks and to detect the IDS threats using a hybrid model that comprises both signature and anomaly techniques. Thereupon, broader routers summarise their findings into a whitelist and disseminate them to the devices. Hence, a device uses this list to handle the harmless packets and forwards unlisted packets to the border router to consult it about them. The results reported by the authors demonstrate the ability of SVELTE to detect sinkhole attacks with 90% and 100% true positive rates over lossy and lossless networks, respectively. However, the requirement that sending suspicious packets over the network to broader routers constitutes the main shortcomings of SVELTE. The work presented in [44] introduces a hybrid IDS model for sinkhole attack detection that takes into account the requirements of IPv6 over Low-Power Wireless Personal Area Networks (6LoWPAN). The proposed model, named Intrusion detection of SiNkhole attacks on 6LoWPAN for InterneT of ThIngs (INTI), consists of two phases: cluster formation and routine monitoring. In the first phase, the network is divided into several clusters, each of which has its cluster head. Thereupon, in the routing monitoring phase, each cluster head scans the traffic generated and forwarded by its members for potential sinkhole attacks using a customised algorithm that quantifies the reputation of each node. Furthermore, this model defines a cooperative mechanism to isolate malicious nodes from the network. The performance evaluation conducted in this work shows that the proposed model can detect sinkhole attacks at 75% and 92% and achieve a false negative rate of 8% and 28% in mobile and stationary network scenarios, respectively. Nevertheless, reliance on some devices to monitor the activities of other nodes can accelerate the depletion of their batteries, which in turn leads to significant degradation of the network connectivity degree over time. The work presented in [45] adopts the same approach as [44], except that it uses a simpler algorithm based on the Dempster–Shafer theory instead of the sophisticated algorithm in INTI. The impact of this replacement is seen in the form of improvements in various performance metrics such as throughput, packet delivery ratio and normalised overhead.

The success in implementing some IPS functions on IoT devices combined with the rapid development of artificial intelligence approaches paves the way for using these approaches to develop a host-based IPS. The model presented in [46] was designed to enable IoT devices to detect common attacks such as scanning, Transmission Control Protocol (TCP) flooding and User Datagram Protocol (UDP) flooding. This model consists of two autoencoder architectures: the first is trained on labelled datasets; the knowledge gained is then stored and transferred to the second autoencoder architecture, whose main task is to find out the class of unlabelled attacks. The results reported in this paper show the advantages of transfer learning in detecting unlabelled attacks. However, one of the major limitations of this model is the significant latency required to train and then transfer the knowledge to each device. Ref. [47] is another transfer learning-based model developed to transfer knowledge for new devices and as well as new attacks. This model employs Genetic Programming (GP) and uses Routing Protocol for Low-power and Lossy Networks (RPL) to evaluate the prediction accuracy. However, the author of this paper cites the problem of code bloat as the major limitation of his proposal. Federated Deep Learning is another approach that has been utilised to develop IPS. DeepFed [48] is designed to detect attacks on industrial cyber-physical systems by empowering each device with a deep learning model consisting of multiple CNNs, gated recurrent units and MLPs. These models are trained locally by each node, and their weights are routed to a designated server whose main task is to aggregate these weights and then disseminate them back to each node to update its weights accordingly. In this way, each node can know about attacks detected by other nodes. In addition, DeepFed uses a cryptographic asymmetric key mechanism to ensure the confidentiality of the communication between the devices and the server. The results reported in this paper show the superiority of this model compared to some selected models using a self-generated dataset with an overall accuracy of up to 99.2%. Investigating the benefits of supervised and unsupervised federated models in detecting IoT malware has been proposed in [49] using MLP and autoencoder architectures, respectively. This paper assesses the performance of the federated learning approach by developing different test scenarios using the N-BaIoT dataset and concludes that the accuracy of the models lies mainly in the method used by the server to aggregate the weights received from nodes. It also points out the profound impact of the adversarial attacks on the federated learning-based models, especially in the unsupervised scenario. In order to overcome the main weaknesses of the federal and transfer learning approaches, the work presented in [50] combines these two approaches to develop a hybrid model. This work considers the case where two networks that use different feature sets to characterise attacks need to exchange their knowledge. The model of this work uses federated learning as an internal learning technique and transfers learning to exchange knowledge. The intensive evaluation presented in this study shows that the accuracy of the model is strongly influenced by the heterogeneity of the feature sets used by the different networks.

Another design approach that has appeared in some work is to treat traffic as images and/or text and then apply the deep learning models that are typically used to classify these types of data sources to detect the attacks. Refs. [51,52,53,54,55,56] are some examples of those models that handle traffic as images; basically, these models convert binary encoded traffic into images and then process them using an ANN model consisting of a set of convolutional and MLP layers. Although these works show that different attacks have different visual representations, the information leakage that results from converting traffic to images may affect the ability of these models to detect advanced attacks. Ref. [53] are amongst those works that use natural language processing techniques to handle traffic. This work uses the sequence of abnormal operating system calls to identify malicious behaviour. However, in the absence of a benchmarking dataset for these calls, the authors deployed several types of attacks on an IoT device and captured their corresponding system calls. This dataset is then digitised and passed through an n-gram to extract the salient features. The proposed model uses Extreme Gradient Boosting (XGBoost) with Long Short-Term Model (LSTM) to predict the future behaviour from the input features and the logistic regression model to classify the system calls as either normal or abnormal. The performance evaluation in this study shows the high sensitivity of the model to the initial values of the model parameters, which makes it difficult to use the model on devices with heterogeneous requirements, as is the case in typical IoT networks.

As it can be seen upon reviewing the above works, the challenge of meeting all the stringent characteristics (heterogeneous and temporal variations in traffic, ad hoc topology, and resource constraints) of IoT networks in a single IDS system has led to trade-offs amongst them. This, in turn, leads to enormous IDS systems that are specifically tailored to particular scenarios. However, the model presented in this paper differs from other works mainly in that it considers all of the above characteristics simultaneously. In particular, our proposal equips each IoT node with an AI-based model that can be used to autonomously detect intrusions without having to exchange their knowledge with other nodes, as is the case with transfer-learning-based models (e.g., [46,47]), or rely on coordinators to adjust the parameters of the node models, as is the case with federated-learning-based models (e.g., [48,49,50]). This makes our proposal more suitable for the decentralised nature of IoT networks, and, more importantly, it avoids exposing sensitive information across the network, as in [43], and saves the resources required to exchange this information. Moreover, our model uses the raw packets as input to the model, so it does not need to apply pre-processing techniques similar to the work presented in [41,42] or conversational mechanisms, as in [51,52,53,54,55,56]. This allows our model to fully consider the content presented in the packets without incurring any overhead and increases the prediction accuracy. Furthermore, by using the self-evolving approach, our proposal is suitable for resource-constrained devices and for heterogeneous and temporal variations in the traffic between nodes. To the best of our knowledge, this is the first work that proposes a self-evolving IDS system for IoT networks.

## 3. Datasets

The accuracy of a proposed intrusion detection/prevention systems is usually determined by evaluating their capabilities to identify different types of attacks provided in one or more benchmark datasets. Therefore, several institutions and research centres have endeavoured to provide enormous datasets that vary in terms of size, availability, recording environment, types of attacks and coverage of protocols, etc. From the multitude of available datasets, this study sets out two conditions for the selection of benchmark datasets: firstly, the selected dataset must contain modern attack types that have taken place in IoT networks. Second, the selected datasets must have been generated from a real network and be provided in the format that can be used to reproduce them at the raw packet level. The key advantage of imposing these two conditions is that they allow a thorough evaluation of our proposal against real-world scenarios. Accordingly, this study uses the following three datasets: BoT-IoT [31], TON-IoT [32], and IoTID20 [33]. Figure 1 illustrates the distributions of classes and subclasses of these datasets, whereas brief descriptions for them are given in the remaining of this section.

The BoT-IoT [31] dataset was created at the IoT lab of the University of New South Wales (UNSW) at Canberra using a realistic network environment. This dataset comprises 72 million records of benign and different cyberattacks, including Denial of Service (DoS), Distributed Denial of Service (DDOS), reconnaissance, and ransomware. The captured raw packets that are generated from this network are made available in the pcap field format of size 16.7 Gigabits. In addition to the pcap files, the UNSW provides processed versions of the IoT-BoT dataset in two formats: firstly, the argus format in which the packets are grouped into flows based on a vector of features, and secondly, the Comma Separated Values (CSV) format which contains the packets’ features and their corresponding classes.

The second dataset used in this study is TON-IoT [32], which stands for Telemetry datasets of IoT services, Operating systems datasets of Windows and Linux, as well as datasets of the Network. TON-IoT was introduced by the creators of the BoT-IoT dataset with the aim of providing a more comprehensive dataset, including normal and various types of attacks that threaten the industrial IoT (IIOT). TON-IoT is generated from a real testbed that includes some modern technologies such as Fog, Edge, and multiple clouding layers. This allows the TON-IoT to encompass the rudimentary attacks in conjunction with campaign correlation hacking that takes place in a heterogeneous environment with both IoT and IIoT. The TON-IoT repository contains 22,339,021 records available in the pcap, Argus, and CSV formats.

IoTID20 [33] is the third dataset used in this study to evaluate the performance of the proposed model. This dataset was created by constructing a smart home network that includes multiple IoT devices connected to the Internet through a border router. Interestingly, this dataset employs one of the widely used botnet infrastructures (i.e., the Mirai botnet) to develop various types of attacks, including host discovery, Telnet brute force, UDP flooding, ACK flooding, and HTTP flooding. The number of records in the IoTID20 dataset is 2,985,994, distributed amongst benign and six main classes of attacks.

## 4. Methods

The main contribution of this study is to devise a lightweight yet effective scheme via which the intrusions can be detected by each IoT device autonomously. The proposed scheme is dubbed (SEHIDS) Self Evolving Host-based Intrusion Detection System as it allows each node to build up its own ANN model based on its performance concerning the characteristics of the received traffic. Interestingly, SEHIDS utilises the built-in loss function of ANN to trigger the evolving process; this, in turn, allows SEHIDS to be used with different sorts of architectures. This study exploits this capability to develop the two common IDS systems: signature-based IDS using MLP architecture and anomaly-based IDS using Replicator Neural Network (ReNN) architecture.

In essence, an MLP is a computational directed acyclic graph that has been developed to imitate the Biological nervous system. The building block of MLP is the artificial neuron which consists of the vector of learnable parameters and an activation function that is used to inhibit or excite these parameters with the aim of approaching the optimal mapping of the inputs to their corresponding outputs. The output is defined in accordance with the purpose for which the MLP model is created; when MLP is used as a classifier, the predefined class of the given input is used as the output, and while when MLP is used as a regressor, the successor instance of the input is used instead. By providing the ground truth to the MLP, it is commonly used as a supervised learning architecture. A typical MLP comprises an input, zero or more hidden, and an output layer where each layer comprises a set of neurons. The open literature is populated with plenty of studies that have utilised ANNs to solve sophisticated problems in vast and diverse fields [17,18,19]. Interestingly, the great competency exhibited by the ANN is justified by the universal approximation theory [22].

The ReNN architecture [30] is similar to MLP in that it consists of a collection of learnable artificial neurons, which are arranged in input, hidden, and output layers. However, ReNN is engineered in such a way that allows it to capture the common patterns embodied in the input and then use this captured knowledge to reproduce the input at the output layer. This requires setting the dimensionality of both input and output layers equal. Since this reproduction does not require providing ReNN with ground truth, it is naturally used as unsupervised learning architecture. Instigated by the simple architecture of ReNN, it has been used as an anomaly detector in numerous disciplines, including intrusions in communication systems [29,57], malignant development in breast cancer, atypical particles in the ionosphere [58], unorthodox biomedical features, aberrate skin lesions, credit card fraud, and segmental speech recognition [59,60].

Despite the differences between the various architectures, the conventional method used to design most current ANN models is to create a bunch of models with different configurations, evaluate their performance using one or more training datasets, and put the model with the best result into practice. Once deployed, this model remains fixed, regardless of its performance in response to real field data. This method is based on three assumptions: firstly, the abundance of devices ‘resources on which these models are deployed, so that no trade-off between accuracy and the resources required by the model is necessary. Secondly, the availability of well-representative training data against which the hypothesis space of the model can be tuned before putting it into practice. Finally, the possibility of applying one or more pruning techniques (e.g., dropping out some layers or skipping some connections of layers) when generalisation problems arise. However, these assumptions are invalid in the IoT network domain due to turbulent traffic, lack of a priori training datasets and scarcity of resources. This, in turn, raises the desperate need to use a more pragmatic approach where each layer is added to perform a specific task, and its overhead is weighed against the benefits it can bring to the model. Consideration of these constraints has led us to apply the principles of constructive neural networks [28] to develop our proposal.

The constructive neural networks differ from the traditional design approach in that it develops the architecture of the model during the learning process. More specifically, the constructive neural networks attempt to acclimatise the model architecture to the problem at hand by starting with a simple architecture in a typical scenario with only one input layer and one output layer. This initial version is trained using real-time datasets, and its performance is monitored. When performance falls below a certain threshold, the constructive neural networks add one or more new layer(s) to the model to increase its computational power. Thusly, the model grows up dynamically in coinciding with the real-time data, which in turn accelerates the convergence of the model towards the optimal approximation region. It is worth noting that some researchers claim [28] that the gradual evolution of the model in conjunction with the narrow parameter space allows it to approach zero classification errors for any finite and non-contradictory data set. Moreover, some works investigating the performance of constructive neural networks conclude that they can solve any learnable problem [61,62,63] in polynomial time. In contrast, some works [64,65,66] have shown that the weight tuning of ANN models developed based on backpropagation learning approaches is not completed in polynomial time, even when these models involve only three neurons. Motivated by these advantages, a multitude of schemes based on the constructive neural networks have appeared in the open literature. A systemic review of these schemes can be found in [27,67] and their references thereof.

However, to the best of our knowledge, none of these schemes has been devised to suit the characteristics of IoT networks. Therefore, one of the main contributions of this study is to use the principles of constructive neural networks to develop a Self-Evolving Host-based Intrusion Detection System (SEHIDS). To elegantly demonstrate this contribution, the problem formulations and high-level abstractions for SEHIDS are presented in Section 4.1, while the detailed description of our proposal and its pseudocode are given in Section 4.2. In order to keep these two subsections concise, they focus on the MLP architectures, which are used as a supervised classifier, whereas the modification(s) required for the ReNN, which is used as an unsupervised anomaly detector, is given in Section 4.3.

### 4.1. Problem Formulation and High Level Abstraction for the Proposed Model

This study considers an IoT network comprising a set of battery-powered devices with wireless communication capabilities that are spread over a given area to form a multi-hop network. No further assumptions or restrictions regarding the operational modes or communication protocols are imposed here; instead, our concentration is on the traffic as seen by each node within the networks. Here, we use S to denote the set of all nodes and 𝓈 to refer to any arbitrary device 𝓈∈𝒮; accordingly let $\xi_{t}$ be a vector comprising the octets of a packet received by 𝓈 at the time instant t. In this instant, the SEHIDS model consists of a sequence of layers numbered from 1 to $L_{t}$, each of which is equipped with several artificial neurons whose number, input, learnable parameters (i.e., weights and biases) and output are denoted by $n_{t}^{\left(l\right)}$,$\;x_{t}^{\left(l\right)}\;$, $w_{t}^{\left(l\right)}$ and $y_{t}^{\left(l\right)}$ respectively, where $1\;\leq l\leq L_{t}$. Furthermore, the $f_{t}^{\left(l\right)}$ is used to denote the activation function employed by the lth layer to map its input $x_{t}^{\left(l\right)}$ to the corresponding output $y_{t}^{\left(l\right)}$ in terms its parameters $w_{t}^{\left(l\right)}$, i.e., $y_{t}^{\left(l\right)}=f_{t}^{\left(l\right)}\left(x_{t}^{\left(l\right)}|w_{t}^{\left(l\right)}\right)$. The input of the first layer $x_{t}^{\left(1\right)}$ is $x_{t}^{\left(1\right)}$ i.e., $x_{t}^{\left(1\right)}=\xi_{t}$ and the output of the last layer $y_{t}^{\left(L\right)}$ is the class of this traffic which can be expressed as:(1)$$y_{t}^{\left(L\right)\;}=\left[\mathrm{Pr}\left(c|x_{t}^{\left(1\right)}\right)\right];\;\forall c\in C$$
where C is a vector of all considered classes; hence, the $y_{t}^{\left(L\right)\;}$ can be written as the result of a sequence of composite functions starting at the $L_{t}$th layer, passing all hidden layers and ending at the 1st (i.e., input) layer, i.e.,
(2)$$y_{t}^{\left(L\right)\;}=f_{t}^{\left(L\right)}\left(x_{t}^{\left(L\right)}|w_{t}^{\left(L\right)}\right)\circ f_{t}^{\left(L-1\right)}\left(x_{t}^{\left(L-1\right)}|w_{t}^{\left(L-1\right)}\right)\circ \ldots \ldots \circ f_{t}^{\left(1\right)}\left(x_{t}^{\left(1\right)}|w_{t}^{\left(1\right)}\right)$$

The distinguishing feature of SEHIDS is that it is not of a fixed architecture; rather, it is evolved by each node based on its prediction performance in response to the traffic it receives. Specifically, a node starts with the most compact form of the SEHIDS model (i.e., a model with an input layer and an output layer, $L_{1}=2$) and then a new hidden layer is added and trained as the performance of the model degrades.

Realising this approach requires specifying the criteria for the evolution process, including the conditions under which a new layer is added, the number of neurons in the new layer, the pattern used to connect this new layer to the current architecture and the learning scheme used to train the model. Interestingly, the stringent characteristics of IoT networks pose great challenges to the design space of these criteria. More specifically, using lenient conditions for adding a new layer can expand the hypothesis space of SEHIDS beyond the reasonable limit, which in turn leads to larger generalisation errors. In contrast, employing harsh conditions for adding a new layer to SEHIDS can reduce its learning capacity leading to biases between different classes. Similarly, adding a new layer with an inappropriate number of neurons negatively affects the performance of SEHIDS. Increasing the number of neurons in the new layer can boost the model’s ability to extract new features from the raw data but at the expense of a higher computational budget and resource utilisation. Although appending a new layer with fewer neurons can cut off the redundant cost, it cannot add real benefits to the classification performance of the model. Another important criterion that needs to be carefully specified is the learning scheme used for training SEHIDS during its evolution. Training the entire model after adding a new layer can aid the model in encoding all possible cases in the model’s parameter space, but this may make it less sensitive to the temporal variations in traffic during the node’s lifetime and overburden the resources of the devices.

To overcome the above challenges, several approaches and techniques have been devised to develop SEHIDS. First, the conditions for adding a new layer are set based on the divergence between the errors produced by the model due to the new input and the residual errors produced by the model so far. This makes these conditions flexible for varying the learning capacity with respect to their true outcomes and avoids the shortcomings associated with setting mild and hard conditions for adding a new layer. Second, the SEHIDS can add the new layers as a hidden layer and adjust the number of neurons to the number of remaining parameters to achieve the optimal degrees of freedom. This gives the SEHIDS the tremendous ability to consolidate its acquired knowledge without expanding its hypothesis space beyond what is required or overburdening the resources of the devices. Finally, the principles of the cascade correlation learning algorithm [29] are used as a means to train the model. This algorithm was chosen because it can use the knowledge gained from training one variant of the same model for subsequent variants without having to start the entire training process from scratch or even train the entire layers of the model. This is hugely important for the SEHIDS model, as its structure changes over time and the training patterns are scattered over irregular intervals (e.g., intended traffic is received incrementally over the lifetime of a node). In addition, limiting the training of the model to the newly added layers can reduce the resource requirements of the model, which is also critical for IoT devices.

To visually illustrate this approach, Figure 2 shows the high-level abstraction of the evolution of the SEHIDS architecture. Figure 2a shows the SEHIDS architecture at time t=1, i.e., at the time when the marked node joins the IoT network, at this time, the SEHIDS comprises two layers: Input and Output, with no hidden layer. Figure 2b, on the other hand, shows SEHIDS at time t when a new hidden layer has been added to the SEHIDS architecture.

### 4.2. Self Evolving Host-Based Intrusion Detection System

The proposed SEHIDS comprises two main subprocesses: firstly, initialising subprocesses, which are executed once during the lifetime of a node to set up the initial parameters of the model, and secondly, the PckRsvd subprocesses, which are executed whenever a node receives a new packet. The pseudocode of these subprocesses is shown in Algorithm 1, whereas the detailed descriptions are given below.
**Algorithm****1:** pseudocode of SEHIDS.**Input**: Stream of packets; MTU; C**Output**: Class of each packet**Initialisation**  
$n_{1}^{\left(1\right)}\leftarrow MTU;\;n_{1}^{\left(L\right)}\leftarrow C;\;w_{1}^{\left(1\right)}\leftarrow random$
**end****New Packet is received**  
$x_{t}^{\left(1\right)}\leftarrow O\left(Packet\right)$
   Generate $y_{t}^{\left(L\right)}$
   Compute $\epsilon_{t}$ using Equation (3)   **if**
$\epsilon_{t}\geq \epsilon_{t-1}$
**then**     **if**
$\epsilon_{t}\notin \left[CI_{min},\;CI_{max}\right]\;$
**then**
     Compute $n_{t+1}^{\left(L-1\right)}$ using Equation (7)     
$L_{t+1}\leftarrow L_{t}+1$
     
$x_{t+1}^{\left(L-1\right)}\leftarrow y_{t}^{\left(L-1\right)}$
     
$y_{t+1}^{\left(L-1\right)}\leftarrow x_{t+1}^{\left(L\right)}$
     Freeze $w_{t+1}^{\left(L-2\right)}$
     Update $w_{t+1}^{\left(L-1\right)}$ using Equation (6)      Freeze $w_{t+1}^{\left(L-1\right)}$
     Update $w_{t+1}^{\left(L-2\right)}$ using Equation (6)     
**else**
     Update $w_{t}^{\left(l\right)};1\leq l\leq L_{t}$ using Equation (6)   
**end**
  
**end**
**end**

An IoT device upon joining the network starts to run the initalisation subprocess, which aims to construct a simple version of SEHIDS consisting of two layers, input and output, i.e., $L_{1}=2$. The number of neurons in the input layer is set to the number of octets defined in the Maximum Transmission Unit (MTU) of the physical protocol used across the network, i.e.,$\;n_{1}^{\left(1\right)}=O\left(MTU\right)$, where O(·) denotes the function used to count the number of octets. The number of neurons in the output layer is set to the number of the classes that packets are allocated to, i.e., $n_{1}^{\left(L\right)}=\|C\|$**,** and the weight of the first layer is set according to “Xavier” initialisation scheme [68], i.e., to random variables that are distributed according to Gaussian probability distribution with zero mean and $\sqrt{1/O\left(MTU\right)}$ standard deviation. This initialisation scheme was chosen because of its wide applicability in this field. The advantages of the proposed settings are twofold: firstly, it allows SEHIDS to gain insightful information about the packets by examining them deeply at the byte level, and secondly, it makes SEHIDS agnostic to higher communication protocols by operating at the physical layer. This, in turn, extends the portability of the proposed model and makes it possible to mitigate those types of attacks targeting specific types of communication layers.

Once an IoT node has constructed its own SEHIDS initial model, it waits until a new packet is received and then executes the PckReved subprocesses. The first step of these subprocesses is to extract the packet contents at the octets’ levels and feeds them into the input of the first layer, i.e., $x_{t}^{\left(1\right)}=\xi_{t}$ and then proceeds to compute the classes corresponding to this input, $y_{t}^{\left(L\right)}$. Thereafter SEHIDS computes the error of the predicted value, which is denoted by $\epsilon_{t}$, and defined as the differences between the ground truth of the given input and the output of the model due to that given input using the loss function. Amongst the wide variety of the loss functions that are proposed in the open literature, this study employs cross-entropy loss function, which can be defined as [19]:(3)$$\epsilon_{t}=\overset{c=\|C\|}{\underset{c=1}{\sum }}T\left(\xi_{t,c}\right)\log\left(y_{t,c}^{\left(L\right)\;}\right)$$
where $T\left(\xi_{t,c}\right)$ is the value of the cth class of the ground truth fed into the model at the time instant t, and $y_{t,c}^{\left(L\right)\;}$ is the corresponding values generated by SEHIDS. This loss function was used due to its ability to speed up the learning process and its low computational cost, since each dataset instance belongs to only one class and the vector containing $T\left(\xi_{t,c}\right)$ are all zeros except one cell. SEHIDS uses the magnitude of this error to either evolve the architecture of the model or leave it in its current configuration. The core concept on which this decision is made is to conduct a two-tier test: firstly, to test whether or not the prediction error of the current input accelerates the convergence of the residual error to zero. If this convergence is achieved, it means that the architecture of the current model is able to competently process the received traffic, so no change is required. If, on the other hand, the recently added error causes the residual error to deviate from zero, this means that the model is trapped in suboptimal regions, and thereby, SEHIDS must evolve the model to leave this region. The condition that must be met to trigger evolution is that the error in time instance t is greater than or equal to the error in the previous instance t−1, i.e., $\epsilon_{t}\geq \epsilon_{t-1}$. Enforcing this condition ensures that the residual error of the SEHIDS model is strictly monotonically decreasing and bounded below for different architectures. These are the necessary and sufficient conditions to guarantee the convergence of the new and residual error according to the Cauchy criteria [69].

The second tier of testing conducted by SEHIDS is to determine whether or not the error due to the current prediction falls within a 95% confidence interval with respect to the prediction errors generated from the model hitherto. Falling outside this range implies that considerable traffic fluctuations beyond the hypothesis space of the model have occurred, i.e., the number of layers in the current architecture is not coincident with the underlying stochastic processes generating the traffic. Hence, the best remediation for this case is to recruit a new hidden layer to aid a node in distinguishing between the errors generated from the transit (i.e., random) and permanent (i.e., systemic) fluctuations. For the sake of accuracy, this study employs the standard Z-value [70] to find the minimum and maximum values of a 95% confidence interval of the residual error which denoted by $CI_{min}$ and $CI_{max}$ and computed as shown in Equations (4) and (5), respectively.
(4)$$CI_{min}=E\left(\epsilon_{\tau }\right)-1.96\;\frac{\sigma \left(\epsilon_{\tau }\right)}{\sqrt{\left\|\epsilon_{\tau }\right\|}};1\leq \tau \leq t$$
(5)$$CI_{max}=E\left(\epsilon_{\tau }\right)-1.96\;\frac{\sigma \left(\epsilon_{\tau }\right)}{\sqrt{\left\|\epsilon_{\tau }\right\|}};1\leq \tau \leq t$$
where $E\left(\epsilon_{\tau }\right)$ and $\sigma \left(\epsilon_{\tau }\right)$ are the arithmetic mean and standard deviation of all errors generated by SEHIDS since it is initalised at t=1 and up to the time instance at which the confidence interval is computed and 1.96 is the Z-value corresponding to the 95% confidence interval. The selection of this computation method is justified by the traffic of an IoT network is resulted from the interaction of several stochastic processes that are temporally correlated; hence computing the confidence interval requires a rigorous algorithm that can consider the mutual relations between readings at different time steps.

If the recent error reading is within a 95% confidence interval, then the SEHIDS realises that there is no massive fluctuation in the traffic and thereby no benefit can be gained from adding a new layer; in such a case, SEHIDS conducts comprehensive training for the current architecture of the model. The main objective of this training is to utilise the output of the loss function, i.e., error, to steer the optimisation algorithm towards the optimal value of the model’s weights in the reverse direction from the output layer toward the input. This optimisation is accomplished by adjusting the weights of all layers proportionally to their contributions towards the error. Depending on the characteristics of the problem at hand and the availability of training datasets beforehand, several approaches have appeared in the open literature. Gradient descent, stochastic gradient descent, nonlinear conjugate gradient, limited-memory Broyden–Fletcher–Goldfarb–Shanno and quick propagation [18,19] are a few to name. Nevertheless, this study considers the limited resources of the IoT devices along with the lack of training datasets that possess the intrinsic characteristics of IoT network’s traffic beforehand and employs the Quick Propagation algorithm (QuickProp) [71] as an optimization algorithm. QuickProp is a second-order optimisation algorithm that utilises a lightweight version of Hessian’s diagonal approach to update each weight in secant steps independently. According to QuickProp the new weights of the lth layer at time instance t, i.e., $w_{t}^{\left(l\right)}$ can be written formally as:(6)$$w_{t}^{\left(l\right)}=w_{t}^{\left(l\right)}+\frac{\nabla_{w_{t}^{\left(l\right)}}ℒ\left({\hat{\xi }}_{t},\xi_{t}\right)}{\nabla_{w_{t}^{\left(l\right)}}ℒ\left({\hat{\xi }}_{t-1},\xi_{t-1}\right)-\nabla_{w_{t}}ℒ\left({\hat{\xi }}_{t},\xi_{t}\right)}\nabla_{w_{t}}ℒ\left({\hat{\xi }}_{t},\xi_{t}\right)$$
where $\nabla_{w_{t}^{\left(l\right)}}ℒ\left({\hat{\xi }}_{t},\xi_{t}\right)$ is the gradient of $w_{t}^{\left(l\right)}$ with respect to the loss function and $\nabla_{w_{t}^{\left(l\right)}}ℒ\left({\hat{\xi }}_{t-1},\xi_{t-1}\right)$ is the gradient at the previous time instance. The justification for the selection of the QuickProp is multifaceted. Firstly, it being a second-order optimisation algorithm allows it to incorporate thorough information about the curvature of the loss function in the vicinity of the current point of the weighting space. This, in turn, expedites approaching the optimal value of the weights [72]. Secondly, QuickProp obviates the need to dictate the learning process based on predefined hyperparameters. Interestingly, most other optimisation approaches, e.g., backpropagation and stochastic gradient descent, require the specification of the step at which the learning process paces down the gradients, called the learning rate. Mis-setting this hyperparameter can result in wasting the device’s resources or overshooting the optimal values. A further justification for selecting the QuickProp is its low computation budget, which results from the fact that each weight is updated independently without having to track changes in the weights using the chain rule, as is the case with backpropagation or stochastic gradient descent. In addition to the aforementioned, QuickProp updates the weights in a multiplicative manner, allows it to suit the large fluctuations in traffic that an IoT device experiences during its lifetime.

On the contrary, when SEHIDS decides that it is necessary to recruit a new layer, it proceeds to determine the number of neurons in this new layer. The main goal of this determination is to extend the hypothesis space of the current SEHIDS architecture so that it can capture the intrinsic properties of the packets well without overburdening the resources of the devices. To achieve this goal, this study uses the Welch–Satterthwaite approximation [73] to calculate the freedom degree of the model (i.e., the degree of freedom is the consummate number of independent variables required to represent the dataset) and then modify the model accordingly. In mathematical parlance, let $\left\|c_{i}\right\|$ is the number of instances of the ith class, $\sigma \left(\epsilon_{t}|c_{i}\right)$ is the standard deviation of the ith class prediction errors and ⌊·⌋ be the floor function. Then the number of new layer’s neurons, denoted by $n_{t+1}^{\left(L-1\right)}$ can be computed by subtracting the Welch–Satterthwaite approximation from the total number of neurons in the current architecture, i.e., $\overset{l=L_{t}}{\underset{l=1}{\sum }}n_{t}^{\left(L\right)}$ which yields:(7)$$n_{t+1}^{\left(L-1\right)}=\lfloor \frac{{\left(\sum_{c_{i}}^{\left\|C\right\|}\frac{\sigma \left(\epsilon_{t}|c_{i}\right)}{\left\|c_{i}\right\|}\right)}^{2}}{\sum_{c_{i}}^{\left\|C\right\|}\left(\frac{1}{\|c_{i}\|-1}\right){\left(\frac{\sigma \left(\epsilon_{t}|c_{i}\right)}{\left\|c_{i}\right\|}\right)}^{2}}\rfloor -\overset{l=L_{t}}{\underset{l=1}{\sum }}n_{t}^{\left(L\right)}$$

The next step after computing the number of neurons in the new layer is to attach it to the current architecture and then train it. Firstly, the number of layers is incremented by one, i.e., $L_{t+1}=L_{t}+1$, and then appending this new layer as the last hidden layer to the current architecture by assigning the input of this new layer to the output of the last hidden layer in the current architecture, i.e., $x_{t+1}^{\left(L-1\right)}=y_{t}^{\left(L-1\right)}$, and the output of the new layer to the input of the output layer in the current architecture, i.e., $y_{t+1}^{\left(L-1\right)}=x_{t}^{\left(L-1\right)}$. Considering the scarce resources of the IoT devices and the need for an effective approach by which the new architecture can move decisively to the optimal regions propound, using the principles of Cascade Correlation (CasCor) [29] as a learning algorithm in this study. The CasCor is based on two principles: firstly, cascading the new layer into the hidden layers one at a time; thereby, the inputs of the new layer (fans-in) are connected to the model’s input layers (if this is the first time that SEHIDS is running) or to the outputs of the hidden layers already existing in the current architecture, whereas the output of the new layer (fans-out) is connected to the output layer of the model. By these means, the depth of the architecture increases gradually, which in turn leads to fostering the learning capacity and hypothesis spaces without a need to affect the flow of information in the input and output layers. The second principle of CasCor is confining the training process to only those links connecting the new layer to the relevant layers; hence no comprehensive training process for the complete model is performed. Furthermore, the CasCor preserves the stability of the model by splitting the training over two phases, each of which is concerned with a single side of the new layer’s connections, i.e., the fans and fans-out of the new layer. This study employs the quick propagation algorithm (QuickProp) described in Equation (6) to train connections of the new layer so that it combines the advantages of both QuickProp and CasCor.

### 4.3. From MLP to ReNN

In the previous two subsections, the MLP architecture was used to demonstrate the underlying concepts of the SEHIDS system and the evolving mechanism. This subsection is devoted to discussing the modifications required to be made in order to use ReNN in lieu of MLP. The first modification is related to the values of the initialisation sub-process parameters, which set the number of classes to 2, i.e., ‖C‖=2 instead of the number of classes defined by the corresponding dataset and the number of neurons in the input layers equally to the number of neurons in the output layer, i.e., $n_{1}^{\left(L\right)}=n_{1}^{\left(1\right)}$. This is because ReNN is used to distinguish normal traffic from anomalies, which implies that the maximum number of groups is always 2, whereas setting the number of neurons in the output layer in such a manner is to ensure that ReNN can reproduce the input at the output layer that is dimensionally identical, which in turn facilitates the comparison of the other inputs. The second modification is related to the loss function, while the MLP architecture uses cross-entropy to determine the extent to which the classes generated by the model deviate from ground truth; in the ReNN architecture, there are neither classes nor ground truth. Conversely, the ReNN uses the mean squared error as the loss function to measure the average magnitude of the errors between the reproduced and the genuine version of the given input, i.e., $\epsilon_{t}=\|\xi_{t}-y_{t}^{\left(L\right)}\|{}_{2}$. The main reason for using this function is its ability to penalise larger errors more heavily, which in turn makes the model more sensitive in detecting anomalies.

It is noteworthy that all the above modifications are due to the differences in the architectures between the MLP and ReNN, and that no changes need to be made for the evolving mechanism given in the pseudocode shown in Algorithm 1. This is because this mechanism does not define its own measure space to trigger the evolving process of the model. Instead, the evolving criteria are specified based on the outcomes of the loss function, which is an essential part of any learning algorithm (whether it is supervised, semi-supervised, or even unsupervised). From a deeper perspective, the evolving machine presented in this study can be viewed as a concept drift scheme that adopts the model architecture to minimise the errors of the loss function resulting from fluctuation of the learning environment. The use of this approach to devise evolving schemes for learning-based algorithms is not uncommon. The work presented in [74], for instance, uses a similar approach to evolve a simple version of Spiking Neural Networks and use it as an unsupervised anomaly detector. Another example is the unsupervised Growing Hierarchical Self-Organising Maps, which extend the fixed architecture of unsupervised self-organising maps by measuring the output error and then using it to determine the evolving orientation of the map [75].

Table 1 summarises the main differences between the initial versions of the MLP and the ReNN.

## 5. Results and Discussion

This section presents the results of the evaluations carried out to assess the performance of the SEHIDS and discusses these results. Section 5.1 summarises the performance metrics used in these assessments, Section 5.2 provides the classification performance results, while the assessments for the evolving process and the resource requirements are given in Section 5.3 and Section 5.4, respectively. Finally, Section 5.5 provides an assessment for using SEHIDS to evolve an anomaly detector.

### 5.1. Performance Metrics

In this study, the following standard statistical metrics are used to quantify the performance of the proposed model: TP, FP, TN FN, accuracy, precision, recall, and F1-score, where the first four metrics stand for the overall True Positive, False Positive, True Negative, and False Negative exhibited by model concerning all classes, respectively, i.e., the TP is computed as $TP=\overset{\left\|C\right\|}{\underset{1}{\sum }}TP_{c_{i}}$, where $TP_{c_{i}}$ is the sum of all cases that belong to the ith class and are classified by the model correctly in agreement with their corresponding ground truth. Similarly, $TN_{c_{i}}$ counts all cases that are classified by the model not in the ith class, and they are truly not in this class, hence $TN=\overset{\left\|C\right\|}{\underset{1}{\sum }}TN_{c_{i}}$. The $FP_{c_{i}}$ and $FN_{c_{i}}$ sum up the number of cases belonging to the ith class that our model classifies them in opposition to their real classes, either by putting them in different classes or putting the instances of other classes in the ith class, respectively. Based on the above, the latter four metrics can be defined as:(8)$$Accuracy=\frac{TP+TN}{TP+TN+FP+FN}$$
(9)$$\mathrm{precision}\;=\frac{TP}{TP+FP}$$
(10)$$\mathrm{recall}=\frac{TP}{TP+FN}$$
(11)$$F1\text{-}\mathrm{score}\;=\frac{2TP}{2TP+FP+FN}$$

In addition to the above statistical metrics, this study employs the following measurements to quantify the resources utilisation demanded by the proposed model: FLoating point OPerations per Second (FLOPS) which counts the number of floating points executed by the model to obtain the results per second and memory footprint in KiloBytes. The assessment presented in this paper used the specifications of the OpenMote-B board [76] as a baseline to measure the performance of SEHID. OpenMote-B is empowered with an ARM Cortex-M3 microcontroller running at 32 MHz with 32 kB RAM and 512 kB internal Flash. This device is powered by 2xAA batteries.

The proposed algorithm, as shown in the pseudocode in Figure 3, was implemented on Arm Development Studio [77], a C/C++ IDE and embedded toolchain platform that was built to allow developers to code and validate their proposals before deploying them on microcontrollers. In addition, our implementation uses the Cortex^®^ Microcontroller Software Interface Standard Neural Network (CMSIS-NN) library [78], which provides a collection of resource-efficient neural network kernels that can be invoked via standard header files and an application programming interface (APIs). These tools were chosen for evaluation because they can provide an environment that closely resembles real IoT devices.

In this study, SEIDS is used to develop the two common types of IDS systems: signature-based and anomaly-based. Since these two types differ in the way they learn and operate, two different training strategies are used here: Online and Batch. The online strategy is used with the signature-based SEIDS models. In this strategy, all the ground truths of three datasets are extracted. Then a sample of the dataset under assessment is randomly selected and then fed into the SEIDS model, which is responsible for processing this dataset. Once the model has generated its result, i.e., the predicted class corresponding to this sample, the ground truth of this sample is passed to the SEIDS to perform the evolving procedures. Once the model has seen all the samples in this dataset, the TP, FP, TN, and FN are calculated for each class as described above and then used to create the confusion matrix. Batch training, on the other hand, is used for the anomaly-based SEIDS models. In this evaluation, the model is trained in an unsupervised manner on benign traffic and then tested on a mixture of benign and malicious traffic using a 10-fold cross-validation strategy.

### 5.2. Assessment of the Classification Performance

The first assessment of the proposed model is carried out by evaluating its classification performance under the three datasets described in Section 3, namely BoT-IoT [31], TON-IOT [32], and IoTID20 [33]. For the purpose of this assessment, three SEHIDS models have been constructed, each of which has been fed by one of these datasets, the outputs of these models are then collected and analysed using standard statistical methods from which the confusion matrices are given in Figure 3.

The results shown in Figure 3 demonstrate that SEHIDS is able to accurately classify the different types of attacks in the three datasets. The lowest reported value is 0.99, while the highest value is 1. This outstanding performance is attributed to the fact that the design philosophies underlying SEHIDS endow it with tremendous capabilities to adapt its architectures in accordance with the traffic it receives. In particular, exploiting the divergence between the errors generated by the model due to the new input and the residual errors previously generated by the model enables SEHIDS to remediate its classification errors instantaneously. More importantly, the use of these constraints facilitates maintaining SEHIDS to its near-minimal architecture, which in turn improves SEHIDS’ ability to capture the intrinsic differences between different class types and embed them error-free in the small hypothesis space. Moreover, the ability of SEHIDS to determine the number of neurons in the new layer as the optimal complement of degrees of freedom leads to an acceleration of the classification performance of the model in a single step. Another outstanding design philosophy that contributes to these excellent results is the use of Quick Propagation and Cascade Correlation as training schemes. These schemes allow the SEHIDS to consolidate the knowledge gained during its development and quickly share this knowledge with the newly added neurons without compromising classification performance.

A further assessment of the overall accuracy of SEHIDS is performed by measuring the normalised error as a function of time. The results of this assessment, as depicted in Figure 4, show that the normalised errors of the three curves drop sharply after a short time and continue in this way until they reach the zero-error regions. These results are not only consistent with the results of the first assessment but also complement them in the sense that SEHIDS does not take too long to reach the zero-classification error, which in turn allows SEHIDS to effectively recognise different types of classes effectively. It is worth noting that the above results are also consistent with the findings of work investigating the behaviour of models that use constructive learning approaches as a framework. These works show that such models are able to approach zero classification error within polynomial time [25,49,50,51].

### 5.3. Assessment of the Evolving Process

Although the previous two evaluations show that SEHIDS can achieve high classification accuracy on different datasets, it is interesting to investigate how SEHIDS can achieve these results. To satisfy this interest, this section traces in detail the evolutionary process that each SEHIDS instance undergoes in response to the three datasets considered above. The number of hidden layers and the number of their neurons, as a function of time, of the three SEHIDS instances, are shown in Figure 5.

As can be seen from these results, all three architectures have zero hidden layers at time t=1 (i.e., the time instance at which the three SEHIDS architectures are initialised); however, as time passes, the different architectures evolve differently. For example, it can be seen that the number of hidden layers of the SEHIDS architectures processing the TON-IoT dataset increases faster than the other two architectures. This behaviour is due to the fact that the number of classes in the TON-IoT dataset is higher than the number of classes in the other two datasets, which in turn requires a deeper architecture to compartmentalise them in an unchallengeable way. Moreover, it is evident from these results that some of the hidden layers in the SEHIDS architectures used to process the TON-IoT have only a few neurons, while other hidden layers include a large number of neurons. This characteristic can be attributed to the fact that the TON-IoT dataset comprises different types of attacks, some of which have intelligible patterns, while other attack types have complicated patterns. In particular, about 31% of the entire TON-IoT dataset belongs to scanning attacks, which are essentially based on sending numerous messages with almost identical content. In a typical scenario, the attacker attempts to scan victims’ open ports by sending multiple probing packets, each targeting a specific port. Consistent patterns of packets facilitate their classification by SEHIDS once a sufficient number of packets have already been already assigned to this class, without the need to use a large number of neurons. It is interesting to note that the above discussion highlights the advantages that the SEHIDS model reaps from using the content of raw packets as its input. Inasmuch as this enables SEHIDS to recognise almost all possible combinations that occur in a type of attack at a fine-grained level and to embed their features firmly in its parameter space, when this advantage is considered in conjunction with the fact that the second-largest portion of the TON-IoT dataset belongs to Distributed Denial of Service (DDoS) attacks targeting the three common ports: HTTP, UDP, and TCP allow us to deduce that the deeper hidden layers of the SEHIDS model processing TON-IoT were developed with a small number of neurons to detect these types of attacks.

The results in Figure 6 also show that although the depth of the SEHIDS used as IDS for the BoT-IoT datasets is comparable to the SEHIDS acting as IDS for the TON-IoT datasets, the number of neurons in some layers is higher in the latter case than in the formers. An interpretation of this difference can be gained by considering that the majority of the records in the BoT-IoT dataset belong to DoS attacks, either centralised or distributed (i.e., about 97.5% in the IoT-BoT dataset compared to 42% in TON-IoT). The strategy on which the DoS attacks are based is to flood the victim’s selected service port(s) with a large number of requests to make these services inaccessible to the intended users. Since these attacks involve packets targeting the same port but containing different requests, these packets have similar headers but different payloads. To detect these types of latent relationships, SEHIDS needs to add more neurons in some layers, which in turn makes some layers in the IoT-BoT’ SEHIDS model denser than their counterparts in the TON-IoT ‘SEHIDS model.

Further support for this interpretation is provided by comparing the depth and density of the SEHIDS model processing the IoTID20 dataset with the SEHIDS model used as IPS for the BoT-IoT dataset. This comparison shows that some hidden layers in the IoTID20-based model are shallower and sparser than the corresponding layers in the IoT-BoT-based model. These differences can be attributed to the fact that the majority of the records in this dataset are of protocol-based DoS attacks. This type of attack aims to exhaust the resources of IoT devices by abusing the mechanisms of the TCP/IP protocol, e.g., by initialising huge TCP connections but not terminating them, as is the case with the SYN TCP flood attack, or by sending a sequence of HTTP GET/POST requests to reach the maximum number of concurrent connections specified by the devices. The structure of the packets belonging to these types of attacks has common characteristics, which in turn allow the SEHIDS to classify them easily.

The evolving behaviour of SEHIDS, which serves as IDS for the IoTID20, shows some interesting results. For example, it can be seen that the number of layers increases rapidly within a short interval and then remains constant before increasing again. This behaviour can be attributed to the fact that this dataset contains several types of attacks, namely: host discovery, Telnet bruteforce, UDP flooding, ACK flooding, and HTTP flooding, which are devised based on the Mirai botnet infrastructure. To enable the accurate classification of these attacks, SEHIDS adds further hidden layers to its architectures to reveal the latent relationships that connect each type of these attacks to the Mirai Botnet. In particular, each of these attacks is preceded by the formulation of the Mirai botnet, which in turn takes place in several stages. In the first stage, the botmaster (the attack) launches the discovery process to find vulnerable IoT devices. Once these devices are located, the botmaster attempts to compromise their credentials and populate the Command and Communication (C2) servers with the compromised devices’ information. The botmaster then uses this information to infect the devices with malware that allows the C2 server to dictate to the bots (malicious devices) to carry out certain attacks simultaneously. During these phases, some known services and ports are used: In the discovery phase, talent services are typically used across all ports, e.g., 23, 2323, 7547, 5555, 23,231, and 37,777, TCP and UDP ports 7547 and 555 are the common ports used to send the malware, and port 48,101 is used to facilitate communication between bots and botmasters. If we look at the Mirai formulation procedures and their port numbers, we can see the main reasons why some layers in the SEHIDS are much denser than others. In particular, we see that the SEHIDS needs fewer neurons to classify the types of Mirai-based attacks that use the same ports for botnet formation as the other attacks.

### 5.4. Assessment of the Resources Requirments

The evaluation of the resources required by SEHIDS models is one of the crucial valuations for their suitability for IoT networks. Instigated by this importance, the number of FLOPS and memory usage required by the SEHIDS models during their processing for the IoT-BoT, TON-IOT, and IoTID20 datasets are measured and illustrated in Figure 6a–c, respectively.

The results in Figure 6a show that the maximum number of FLOPS required by the SEHIDS to process the BoT-IoT dataset is around 4000, which is far below the processing power of most microcontrollers used in modern IoT devices. To put things into perspective, the range of low-power microcontrollers that are typically used in these devices have the capability to carry out several megaFLOPS per second [76]. This not only attests to the appropriateness of SEHIDS for low-power networks but also provides an explanation for the FLOPS required by SEHIDS elapsed for a fraction of tiny interval during the lifetime of the devices. Interestingly, this figure shows that the number of FLOPS decreased dramatically and scattered over long intermediate intervals. The main rationale behind this characteristic is the reliance of SEHIDS on cascade correlation as a learning scheme. With this approach, it is not necessary to perform extensive training every time the architecture is evolved. Instead, in the cascade correlation, training is limited to only those connections that are relevant to the newly added layer. This leads to a significant reduction in computational costs during the evolution process of the SEHIDS models. Another plausible reason for the low computational cost of the three SEHIDS models is the use of Quick Propagation as a means of updating the weights. Recall that this scheme adaptively steers the weights to the optimal regions based on the curvature of the loss function. Moreover, the Quick Propagation schemes perform updates for the fan-in and fan-out of each new hidden layer separately. This, in turn, makes the computational cost per time associated with updating the weights low. It is interesting to note that the fluctuations that engulf the FLOPs values are due to the updating of the weights performed by the SEHIDS models whenever their recent errors are within the 95% confidence intervals of the residuals, as described in the pseudocode. A comparison of the FLOPS characteristics of the SEHID models processing the BoT-IoT dataset (Figure 7a) with the FLOPS characteristics of the SEHID models processing IoTID20 and TON-IoT datasets (Figure 6b,c, respectively) shows that the number of FLOPS required for the latter two cases is lower than for the former. An interpretation for this characteristic can be gained by considering that the fewer classes there are in a dataset, the fewer neurons are required to represent them in SEHID and, consequently, lower computational costs. Face validation for the result shown in the above figures can be obtained by computing the number of FLOPS demanded by SEHIDS theoretically during the initialisation phase. According to [79], the number of FLOPS of an MLP network with n neurons in the input layer and m neurons in the output layer is FLOPS=(2∗n−1)∗m. The pseudocode in Figure 3 shows that in the initialisation phase, the number of neurons in the input layer is set to MTU (128 in this assessment), and the output layers of the three SEHIDS models are set to the number of classes of these datasets (17, 11, 10 for BoT-IoT, IoTID20, and TON-IoT, respectively). Substituting these numbers into the said equation yields the same FLOPS number illustrated in Figure 7. it is worth noting that the maximum number of FLOPS is obtained at the initialisation phase, after which a single layer with a few neurons is added, which demands small FLOPs.

The characteristics of RAM utilisation of SEHIDS, shown in Figure 7, demonstrate that SEHIDS has s small memory footprint.

These characteristics are due to the various lightweight techniques used in the development of SEHIDS. In particular, encoding the inputs using 8-bits and the evolution of the model over time saves the amount of memory that would be required to permanently store and manipulate sophisticated model architectures, as is the case with traditional ANN design concepts. In addition, the use of online training techniques, combined with the restriction of training to certain connections (due to the use of Cascade Correlation) and the carrying out weight updates in phases (due to the use of Quick Propagation), reduces the size of the weighting matrices and the reading and writing operations required to update them.

### 5.5. Assessment of SEHIDS with ReNN Architecture (Anomaly-Based IDS)

The results presented in the previous subsections show that SEHID is able to evolve a simple version of the MLP architecture in such a way that allows it to accurately detect different types of attacks without consuming a significant number of resources. However, since the MLP architecture is by far a supervised learning approach, the resulting model acts as a signature-based IDS. This section is dedicated to assessing the ability of SEHID to develop an anomaly-based IDS system.

The results presented in the previous subsections show that SEHIDS is able to evolve a simple version of the MLP architecture in such a way that allows it to accurately detect different types of attacks without consuming a significant number of resources, which attests to its suitability for IoT networks. However, since the MLP architecture is by far a supervised learning approach, the resulting model acts as a signature-based IDS. This section is devoted to assessing the ability of SEHIDS to develop an anomaly-based IDS system. In order to conduct a thorough investigation of the performance of SEHIDS in the domain of anomaly-based IDS, three new SEHIDS models are created, each of which is used with one of the three datasets described in Section 3. The number of neurons in the input and output layers of the three initial ReNN architectures fed into the three SEHIDS models is set to 128, which is the number of MTU’s octets. These models are trained on benign traffic in an unsupervised manner (i.e., without labelling data) using a 10-fold cross-validation strategy. In this strategy, the considered dataset is randomly split into 10 subsets where each subset is used in the training phases while the remaining is used in the testing phase. During the training phase, the SEHIDS is used to evolve the initial architectures until the evolving criteria defined in the pseudocode are met, i.e., until the error calculated by the loss function in a time instance becomes smaller than the errors produced by the model up to that point. Thereafter, the resultant architecture is maintained, SEHIDS is halted, and a mixture of benign and malicious traffic is used to test the performance of the model. During the testing phase, the minimum error obtained during the training phase is used as the confines for normal instances.

The performance of the three SEHIDS-based models is measured using standard statistical metrics as the average metrics that are obtained during the testing phase and then compared with the results of related work. Table 2 summarises these results, where the abbreviations are defined as follows.

LR: Logistic Regression, LDA: Linear Discriminant Analysis, RF: Random Forest, CART- Classification and Regression Trees, NB- Naïve Bayes, CART- Classification and Regression Trees, LSTM- Long Short-Term Memory, AE: autoencoder, iForest: Isolation Forest, OC-SVM: One Class-SVM, CNN: Convolutional neuronal network, GRU: Gated Recurrent Unit, DAGMM: Deep Autoencoder Gaussian Mixture Model, OPTICS: Ordering Points to Identify the Clustering Structure, BiLSTM: Bidirectional, MTL: Multi-Task Learning, IG: Information Gain Ensemble, GRE: Gain Ratio Ensemble, UMF: Union Mathematical set theory FS, LFN: Single Hidden Layer Feed-Forward Neural Network.

The results presented in Table 2 demonstrate the outperformance of SEHIDS compared to the other models, which affirms the ability of SEHIDS to develop anomaly-based IDS with the same efficiency as signature-based IDS. The generic performance is mainly attributed to the fact that SEHIDS does not define its own measure space to trigger the evolving process but uses the outcomes of the built-in loss function to do so. This, in turn, allows SEHIDS to act as a concordant concept-drift scheme through which the model can increase its computational power timely and without compromising its intrinsic operations.

## 6. Conclusions and Future Works

This paper proposed SEHIDS: Self Evolving Host-based Intrusion Detection System as a novel class of ANN lightweight IDS system for IoT networks. The underlying approach of SEHIDS is to provide each IoT device with a small version of an ANN architecture and a resource-friendly mechanism through which the architecture can be trained and evolved as its performance degrades. SEHIDS was proposed to address the desperate need to protect IoT networks from advanced cyber-attacks and was specifically designed to meet their stringent characteristics. This includes retaining the ANN architecture to its near-minimal configurations with the aim of reducing the time and resources required to run it and accelerating the approach to the optimal operation region. In addition, the SEHIDS allows each device to build up its ANN architecture to cope with the threats it faces, which allows SEHIDS to deal with discrepancies and turbulence in traffic amongst nodes. The various advanced approaches and techniques utilised to design SEHIDS facilitate decoupling the simple ANN architecture from the evolving mechanism, which in turn allows us to develop two common types of IDS: the signature-based in which the super-vised architecture, i.e., MLP, is used to classify different types of attacks, and the anomaly-based IDS version, in which an unsupervised architecture, i.e., ReNN, is used to distinguish benign from malicious traffic. The comprehensive assessments demonstrate the high competence of SEHIDS to fulfil the objectives for which it was developed. Indeed, the average achieved true-positive rate is 1. Furthermore, these results show that the resources consumed by SEHIDS are in the order of small fractions of typical IoT devices’ resources.

During the development of this work, several possible future research directions have emerged. One of the most important directions is the use of protocol validation tools that can accelerate the deployment of SEHIDS in a test bed environment. Another important future direction is to extend the functionality of SEHIDS so that it can act as an autonomous host-based IPS.

## Figures and Tables

**Figure 1 sensors-22-06505-f001:**
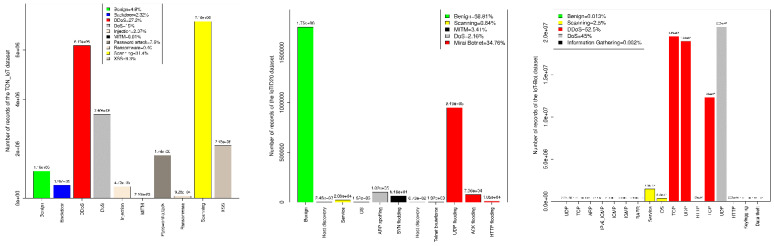
Distributions of classes and subclasses of the three datasets used in this study.

**Figure 2 sensors-22-06505-f002:**
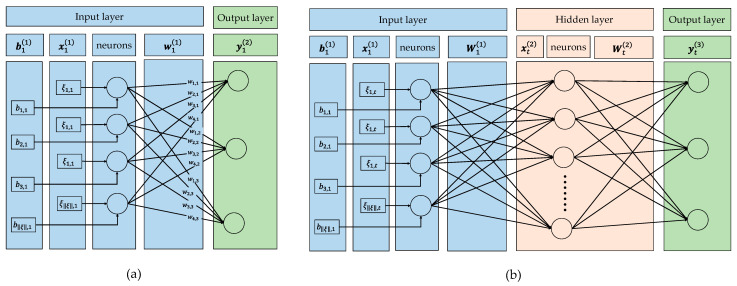
High level abstraction for the evolving of SEHIDS (**a**) at initialisation phase and (**b**) after elapsing some time t.

**Figure 3 sensors-22-06505-f003:**
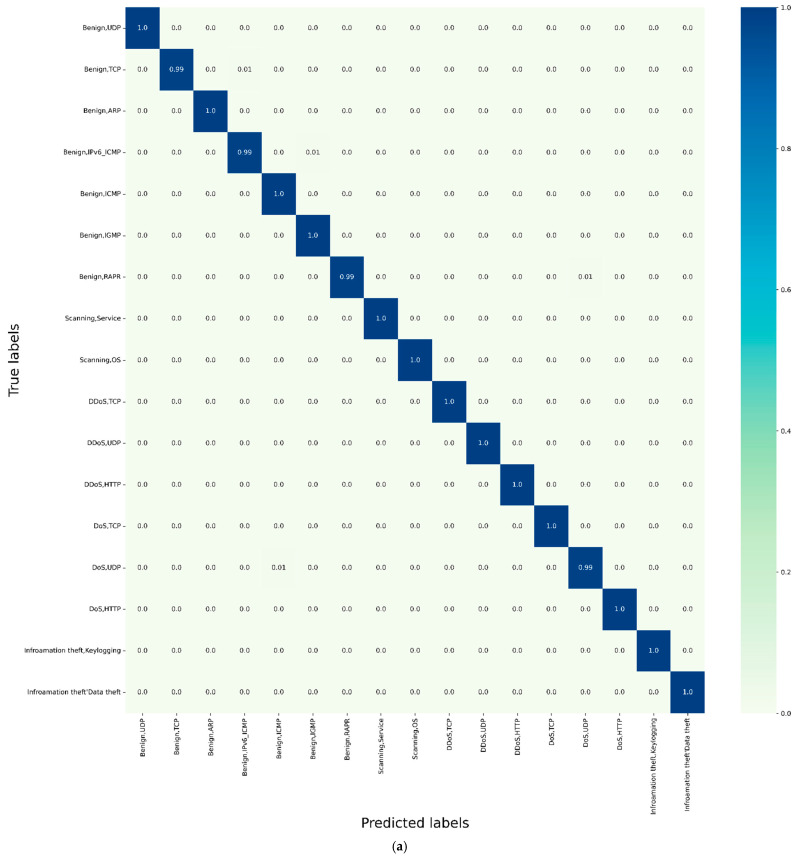
Confusion matrices of (**a**) BoT-IoT, (**b**) IoTID20, and (**c**) TON-IoT.

**Figure 4 sensors-22-06505-f004:**
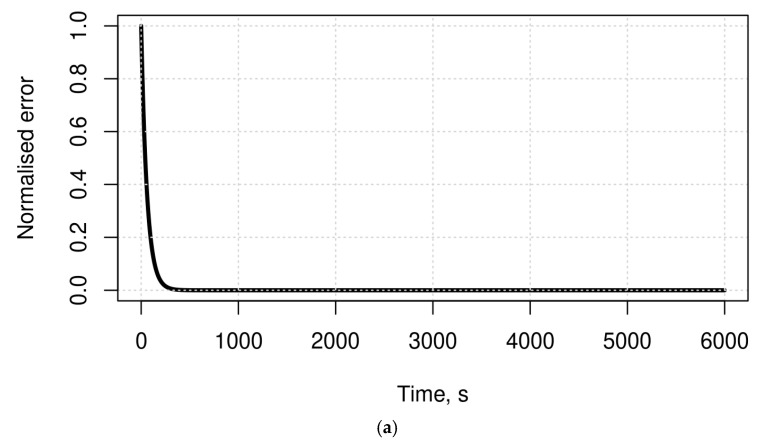
Normalised error versus time of (**a**) BoT-IoT, (**b**) IoTID20, and (**c**) TON-IoT.

**Figure 5 sensors-22-06505-f005:**
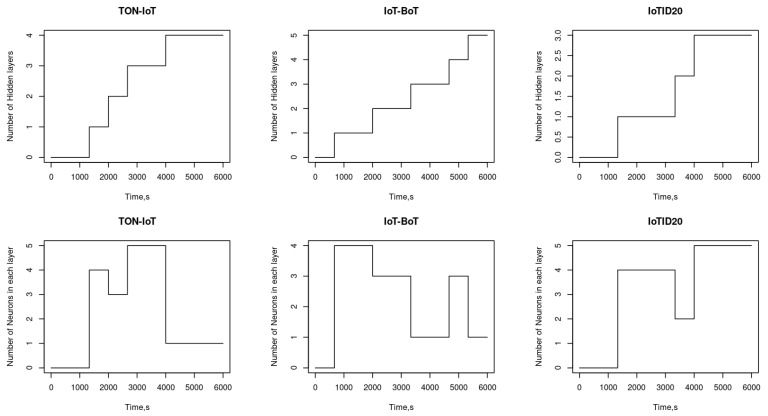
Evolving of SEHIDS for BoT-IoT, IoTID20 and TON-IoT.

**Figure 6 sensors-22-06505-f006:**
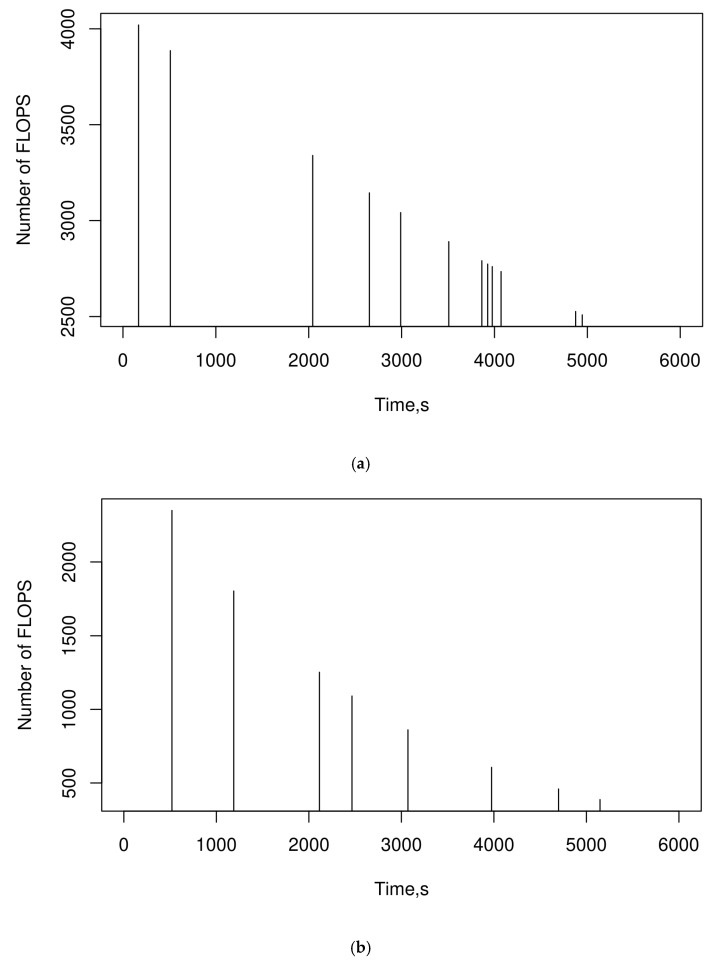
FLOPS of (**a**) BoT-IoT, (**b**) IoTID20, and (**c**) TON-IoT.

**Figure 7 sensors-22-06505-f007:**
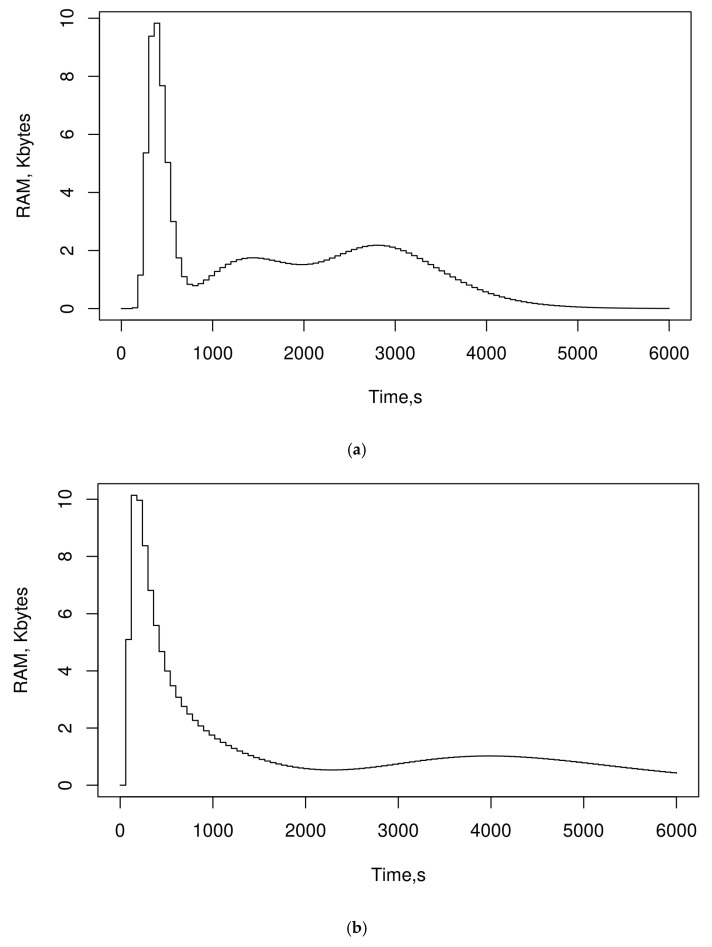
RAM utilisation of (**a**) BoT-IoT, (**b**) IoTID20 and (**c**) TON-IoT.

**Table 1 sensors-22-06505-t001:** Parameters of the initial versions of MLP and ReNN architectures.

Parameter	MLP Architecture	ReNN Architecture
Number of classes	According to the dataset	‖C‖=2
Number of neurons in the input layer	$n_{1}^{\left(1\right)}=O\left(MTU\right)$	$n_{1}^{\left(1\right)}=O\left(MTU\right)$
Number of neurons in the input layer	$n_{1}^{\left(L\right)}=\|C\|$	$n_{1}^{\left(L\right)}=n_{1}^{\left(1\right)}$
Output of neurons in the input layer	$y_{t}^{\left(L\right)\;}=\left[\mathrm{Pr}\left(c|x_{t}^{\left(1\right)}\right)\right];\;\forall c\in C$	$y_{t}^{\left(L\right)\;}={\hat{\xi }}_{t}$
Loss function	$\epsilon_{t}=\overset{c=\|C\|}{\underset{c=1}{\sum }}T\left(\xi_{t,c}\right)\log\left(y_{t,c}^{\left(L\right)\;}\right)$	$\epsilon_{t}=\|\xi_{t}-y_{t}^{\left(L\right)}\|{}_{2}$

**Table 2 sensors-22-06505-t002:** Comparison between the performance metrics of SEHIDS based models and some related works.

Dataset	Ref.	Architecture	Accuracy	Precision	Recall	F1-Score	Training Time (s)	Testing Time (s)
TON_IoT	[80]	LR	0.61	0.37	0.61	0.46	3.023	0.005
	[80]	LDA	0.68	0.74	0.68	0.62	1.041	0.004
	[80]	kNN	0.84	0.85	0.84	0.84	58.018	109.361
	[80]	RF	0.85	0.87	0.85	0.85	10.884	0.164
	[80]	CART	0.88	0.90	0.88	0.88	6.308	0.022
	[80]	NB	0.62	0.63	0.62	0.51	0.21	0.069
	[80]	SVM	0.61	0.37	0.61	0.456	3525.052	558.663
	[80]	LSTM	0.81	0.83	0.81	0.80	1596.809	9.023
	[81]	AE + LSTM + iForest	-	0.7059	0.9001	0.7913	396.317	2.142
	[81]	OC-SVM	-	0.7004	0.4047	0.5130	4659.594	1431.322
	[81]	iForest	-	0.7634	0.3650	0.4939	19.586	5.158
	[81]	LSTM-AE	-	0.6565	0.3025	0.4142	1900.681	55.876
	[81]	ID-CNN-GRU	-	0.4699	0.8442	0.6038	575.203	15.318
	[81]	AE	-	0.625	0.3201	0.4233	97.403	2.465
	[81]	DAGMM	-	0.6264	0.4810	0.5442	263.947	0.173
	[82]	Mini Batch K-Means	0.8772	0.933	0.698	0.799	-	-
	[82]	OPTICS	0.7935	0.633	0.967	0.765	-	-
	[82]	Fuzzy C-Means	0.9105	0.993	0.748	0.854	-	-
	**This work**	**ReNN + SEHIDS**	**0.999**	**0.999**	**0.999**	**0.999**	**2.325**	**7.104**
BoT-IoT	[83]	LSTM	0.9990	0.9995	0.9995	0.9995	-	-
	[83]	BiLSTM	0.9996	0.9997	0.9997	0.9997	-	-
	[83]	GRU	0.9993	0.9997	0.9997	0.9997	-	-
	[84]	RNN-BPTT	0.9820	-			-	-
	[85]	MLP	0.7901	-	-	-	-	-
	[85]	CNN	0.9127	-	-	0.9780	-	-
	[86]	RNN	0.9831	0.9750	-	0.9980	-	-
	[87]	ANN	0.9723	-	-	-	-	-
	[87]	GRU	0.9976	-	-	-	-	-
	[88]	DNN	0.9980	0.9780	-	-	-	-
	[88]	CNN	0.9910	0.9890	-	-	-	-
	[88]	RNN	0.9990	0.9970	-	-	-	-
	[88]	MTL	0.9990	0.9980	-	-	-	-
	[89]	DNN	0.9990	-	-	-	-	-
	**This work**	**ReNN + SEHIDS**	**1.00**	**1.00**	**1.00**	**1.00**	**3.340**	**8.748**
IoTID20	[90]	AE	0.9520	0.9700	-	-	-	-
	[91]	ANN with UMF	0.9907	0.9910	0.9980	0.9900	42.40	-
	[91]	C4.5 with UMF	0.9991	0.9990	0.9990	0.9990	44.18	-
	[91]	Bagging with UMF	0.9991	0.9990	0.9990	0.9990	50.25	-
	[91]	Ensemble with UMF	0.9998	0.9990	0.9990	0.9990	56.75	-
	[33]	SVM	0.4000	0.5500	0.3700	0.1600	-	-
	[33]	Ensemble	0.8700	0.8700	0.8700	0.87	-	-
	[92]	SVM	0.9700	-	-	-	-	-
	[92]	Ensemble	0.9400	-	-	-		-
	[92]	SLFN	0.9800	-	-	-		
	[93]	XGBoost	0.9979	-	0.98	1.00		
	[93]	SVM	0.9876	-	0.98	0.9800		
	**This work**	**ReNN + SEHIDS**	**1.00**	**1.00**	**1.00**	**1.00**	**4.569**	**9.010**

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
