# Peer review of "SEHIDS: Self Evolving Host-Based Intrusion Detection System for IoT Networks"

_sensors, 2022, doi:10.3390/s22176505_

Round 1

Reviewer 1 Report

This work is motivated by the need to protect IoT networks by proposing SEHIPS as a host-based, self-evolving intrusion prevention system for IoT networks.  The main idea is to provide each IoT node with a simple artificial neural network architecture and lightweight mechanisms through which an IoT device can train this architecture online and evolve it whenever its performance prediction degrades.

SEHIPS allows each node to generate the necessary ANN architecture to prevent the threats it faces by tailoring the solution to achieve heterogeneity and traffic turbulence between nodes.

Interestingly, we have conducted evaluations of SEHIPS from different perspectives with three datasets. The results of the evaluations demonstrate that SEHIPS is capable of accurate predictions.

As a complementary work, it is proposed to validate the proposal with a protocol validation tool as in "AVISPA in the validation of Ambient Intelligence Scenarios". Another alternative and complementary proposal is the use of certification for the protection of IoT networks by proposing a scheme that combines hardware (TPM) and software certification as in "Software and Hardware Certification Techniques in a Combined Certification Model".

Reviewer 2 Report

The paper presents SEHIPS, an intrusion detection system which relies on constructive neural networks. Authors detail how they build such an intrusion detector by using a MLP which is periodically updated by adding a new hidden layer whenever specific conditions trigger a re-training of the model. This is very important as IoT systems and ICT systems in general are often meant to evolve during their operational life: therefore, any detection strategy is meant to evolve as well to cope with dynamic changes of the environment.

The paper is well written albeit there is so much text that could be cut out to shorten the paper, especially in Section 1, 2, 4.1 and 4.2.

However, at the current state the paper has the following problems that - according to the reviewer - are showstoppers for accepting this paper for publication.

1) The title says that SEHIPS is an intrusion prevention system. However, this is clearly an intrusion Detection system: there are no means to block an ongoing attack once detected, the ANN only performs detection. The paper should be re-written accordingly.

2) The paper blindly adopts neural networks to perform intrusion detection, but there are recent works (i.e., Shwartz-Ziv, R., & Armon, A. (2022). Tabular data: Deep learning is not all you need. Information Fusion, 81, 84-90) that argue about the usage of DNNs for processing tabular data, saying that adopting classic supervised approaches (extreme gradient boosting XGBoost in that paper) may achieve better or equal classification performance while being faster and generating lighter models (which is a concern in IoT). Also, those models have checkpointing / transfer learning capabilities that could be used to update the model as the system evolves. Therefore authors should explain why they blindly used MLP or at least compare with classical models (e.g., Random Forests, XGBoost, ...)

3) MLP is a supervised algorithm. Therefore to update the model you need labels for your recent data that you use to train the new hidden layer for SEHIPS. This may generate problems in a real environment since devising labels is probably the most difficult and time-consuming activity, that can also hardly be fully automatized. How do you deal with this problem?

4) Which connects to 3). When a system evolves, it may happen that unknown attacks (often called zero-day attacks) are crafted against the system itself. An intrusion detector should be as robust as possible against them. This aspect was completely left out frm the paper. I understand it is difficult to integrate it in your work, but Section 2 hould at least discuss this aspect, als because there are unsupervised approaches that can detect them without needing labels for training (that is why this is connected to concern 3 above). Papers to start such discussions could be

- Palani, K., Holt, E., & Smith, S. (2016, March). Invisible and forgotten: Zero-day blooms in the IoT. In 2016 IEEE international conference on pervasive computing and communication workshops (PerCom Workshops) (pp. 1-6). IEEE.

- Zoppi, T., Ceccarelli, A., & Bondavalli, A. (2021). Unsupervised algorithms to detect zero-day attacks: Strategy and application. Ieee Access, 9, 90603-90615.

- Lobato, A. G. P., Lopez, M. A., Sanz, I. J., Cardenas, A. A., Duarte, O. C. M., & Pujolle, G. (2018, May). An adaptive real-time architecture for zero-day threat detection. In 2018 IEEE international conference on communications (ICC) (pp. 1-6). IEEE.

- Blaise, A., Bouet, M., Conan, V., & Secci, S. (2020). Detection of zero-day attacks: An unsupervised port-based approach. Computer Networks, 180, 107391.
